

# **Technical Note: DACNO<sub>2</sub> – A Multi-Constraint Deep Learning Framework for High-Resolution 3D NO<sub>2</sub> Field Estimation**

Wenfu Sun<sup>1, 2</sup>, Frederik Tack<sup>1</sup>, Lieven Clarisse<sup>2</sup>, Michel Van Roozendael<sup>1</sup>

Correspondence to: Wenfu Sun (wenfu.sun@aeronomie.be)

Abstract. Accurate, high-resolution 3D fields of nitrogen dioxide (NO2) are critical for air quality management and satellite retrievals, yet traditional chemistry-transport models (CTMs) face challenges in fine-scale modeling. Machine learning (ML) alternatives often struggle with generalization and transferability, inheriting biases from CTMs or being limited by sparse surface measurements. We present the Deep Atmospheric Chemistry NO<sub>2</sub> model (DACNO<sub>2</sub>), a deep learning model that generates daily 2 km 3D NO<sub>2</sub> fields over Western Europe. The model's three-phase and multi-constraint training strategy begins by pre-training on European Copernicus Atmosphere Monitoring Service (CAMS) reanalysis data to learn large-scale atmospheric patterns, then fine-tunes with both CAMS and in-situ European Environmental Agency (EEA) surface data to correct biases and refine local detail, and completes with an adaptive fine-tuning to capture evolving trends. An evaluation for 2023 shows that DACNO<sub>2</sub> reproduces broad-scale 3D CAMS fields (R<sup>2</sup> = 0.90) while improving agreement with independent EEA stations over the CAMS reanalysis (R<sup>2</sup> enhanced from 0.61 to 0.66; bias reduced from -1.15 to -0.38 µg/m<sup>3</sup>). The model resolves more spatial detail and learns physically interpretable relationships. This hybrid training approach fuses the physical consistency of a process-based model with the real-world accuracy of surface measurements, overcoming the limitations of using either constraint data alone. Applying DACNO2 a-priori profiles to TROPOMI retrievals increases tropospheric NO<sub>2</sub> columns by 3% on average over those using European CAMS profiles, with larger enhancements over emission hotspots. These results demonstrate the framework's potential to advance air quality monitoring and satellite remote sensing.

<sup>&</sup>lt;sup>1</sup> Royal Belgian Institute for Space Aeronomy (BIRA-IASB), Brussels, Belgium.

<sup>&</sup>lt;sup>2</sup> Université libre de Bruxelles (ULB), Spectroscopy, Quantum Chemistry and Atmospheric Remote Sensing (SQUARES), Brussels, Belgium.




## 25 1. Introduction

Nitrogen dioxide (NO<sub>2</sub>) is a key atmospheric pollutant with significant impacts on air quality, human health, ecosystems, and atmospheric chemistry. Primary sources include traffic, industrial activities, and energy production, with additional contributions from natural emissions (Crippa et al., 2018). Accurate characterization of the spatiotemporal distribution of NO<sub>2</sub> is critical for both air pollution management and atmospheric chemistry research.

Chemistry Transport Models (CTMs) such as GEOS-Chem (Bey et al., 2001), TM5-MP (Krol et al., 2005; Williams et al., 2017; Huijnen et al., 2010), WRF-Chem (Grell et al., 2005), and the Copernicus Atmosphere Monitoring Service (CAMS) (Peuch et al., 2022; Inness et al., 2019) are widely used to simulate atmospheric NO<sub>2</sub> based on physical and chemical processes. However, most CTMs operate at coarse spatial resolution due to computational constraints and the limited availability of high-resolution emission inventories. This restricts their ability to represent fine-scale NO<sub>2</sub> variability and often results in spatial smoothing and underestimation, particularly in urban environments. Emission inventories are usually outdated and may omit localized and small-scale sources (Lu et al., 2025), contributing to uncertainties and discrepancies between bottom-up and top-down emission estimates (Kuik et al., 2018; Yang et al., 2021). While regional high-resolution CTMs are available, such as CAMS at 10 km (Douros et al., 2023; Ialongo et al., 2020) and regional WRF-Chem at 3 km (Kuhn et al., 2024b), challenges remain in accurately capturing urban and fine-scale NO<sub>2</sub> patterns (Meleux et al., 2024), and model optimization is often resource-intensive (Kuhn et al., 2024a; Kuhn et al., 2024b).

CTM outputs also serve as a-priori NO<sub>2</sub> profiles for satellite retrievals (Palmer et al., 2001; Douros et al., 2023; Yang et al., 2023), supporting large-scale NO<sub>2</sub> monitoring. Over the past three decades, satellite NO<sub>2</sub> observations have been advancing toward higher spatiotemporal resolution. Satellite instruments such as the TROPOspheric Monitoring Instrument (TROPOMI, 7 × 3.5 km<sup>2</sup>, 5.5 × 3.5 km<sup>2</sup> since August 2019) on Sentinel-5P (Veefkind et al., 2012), the Geostationary Environment Monitoring Spectrometer (GEMS, 3.5 × 8 km<sup>2</sup>) (Kim et al., 2020), Tropospheric emissions: Monitoring of pollution (TEMPO, 2 × 4.75 km<sup>2</sup>) (Zoogman et al., 2017), Sentinel-4 (8 × 8 km<sup>2</sup>) (Gulde et al., 2017), Sentinel-5 (7.5 × 7.5 km<sup>2</sup>) (Bézy et al., 2014), Twin ANthropogenic Greenhouse Gas Observers (TANGO, 300 × 300 m<sup>2</sup>) (Landgraf et al., 2020), and the Copernicus Anthropogenic Carbon Dioxide Monitoring constellation (CO2M, 2 × 2 km<sup>2</sup>) (Bernd et al., 2021) are advancing spaceborne NO<sub>2</sub> observations to kilometer-scale resolution. This progress has increased demand for high-resolution a priori profiles,

https://doi.org/10.5194/egusphere-2025-4259 Preprint. Discussion started: 24 November 2025

© Author(s) 2025. CC BY 4.0 License.






which can improve the sensitivity of satellite NO<sub>2</sub> products to near-surface concentrations and emission hotspots. However, CTM-based profiles remain constrained by the limitations mentioned above, highlighting the need for alternative modeling approaches.

Machine learning (ML) provides an efficient alternative for high-resolution NO<sub>2</sub> estimation. ML techniques have been widely applied for surface NO<sub>2</sub> mapping (Sun et al., 2024; Kim et al., 2021; Wei et al., 2022), and recent studies have extended these approaches for 3D NO<sub>2</sub> modeling above the surface. These studies have trained models on process-based 3D NO<sub>2</sub> fields generated by CTMs (Bodnar et al., 2024; Kuhn et al., 2024a), on vertical profiles from MAX-DOAS observations (Zhang et al., 2025; Zhang et al., 2022b; Jiang et al., 2025), and on a combination of process-based CTM outputs with satellite observations (Li and Xing, 2024). While these studies demonstrate the feasibility of ML-based 3D NO<sub>2</sub> modeling, challenges remain in achieving high spatial resolution, robust generalization, and transferability. Process-based data carries inherent biases and has relatively coarse resolution. Ground-based observations are sparse and unevenly distributed, limiting the model's spatial generalization. While Li and Xing (2024) combine process-based NO<sub>2</sub> fields with satellite NO<sub>2</sub> observations from the Ozone Monitoring Instrument (OMI) to train the ML model, the resulting product is still limited to a coarse resolution (27 km).

In this study, we present the Deep Atmospheric Chemistry NO<sub>2</sub> model (DACNO<sub>2</sub>), a deep learning model developed to produce daily, high-resolution (2 km) 3D NO<sub>2</sub> fields with high accuracy, robust generalization, and transferability. DACNO<sub>2</sub> integrates multi-source inputs, including emissions, geography, meteorology, and temporal indicators. The model is trained using a phased, multi-constraint approach that combines process-based CAMS fields with ground-based EEA measurements. This method enables the model to reproduce broad-scale, process-based NO<sub>2</sub> patterns and capture local NO<sub>2</sub> gradients. The training strategy consists of three phases: pre-training, multi-constraint fine-tuning, and adaptive fine-tuning. Western Europe (5°W–9°E, 42°N–54°N) is chosen as the study region, given its diverse topography, high urbanization, and substantial industrial activity.

This study addresses two key research questions: (1) Can a deep learning framework combining multi-constraint and phased training overcome the resolution, bias, and generalization limitations of current CTM and ML approaches for 3D NO<sub>2</sub> modeling? (2) Does the DACNO<sub>2</sub> 2 km product improve fine-scale NO<sub>2</sub> representation to support applications in regional air quality management and satellite retrievals?

The remainder of this paper is organized as follows. Section 2 describes the DACNO<sub>2</sub> development framework, including dataset preparation, model architecture, and training strategy. Section 3 evaluates model performance. Section 4 discusses broader insights and implications. Conclusions and outlook are provided in Section 5.




## 2. Development Framework for DACNO2

#### 2.1 Framework Overview

Figure 1. Overview of the DACNO2 model development framework. The framework integrates multiple input data streams: temporal indicators, emission inventories and proxies, geography, and ERA5 meteorological data, with two target datasets: process-based NO2 from CAMS European air quality reanalysis and ground-based in-situ EEA NO2 measurements. The training is organized in three sequential phases: pre-training on process-based CAMS NO2, multiconstraint fine-tuning with both CAMS and EEA data, and adaptive fine-tuning to recent NO2 trends. The resulting model generates daily, high-resolution (2 km) 3D NO2 fields. Arrows indicate the data flow and phased training process.

DACNO<sub>2</sub> is developed to provide daily 3D NO<sub>2</sub> fields at high spatial resolution (2 km) with improved accuracy and generalizability by integrating multi-source data, physically consistent process-based datasets, and real-world measurements. The overall framework, illustrated in Fig. 1, combines diverse data streams with a phased training strategy.

DACNO<sub>2</sub> uses five groups of input features: temporal indicators, emission inventories and proxies, geographic data, ERA5 single-level meteorological variables, and ERA5 multi-level meteorological variables. Together, they provide complementary information on spatial and temporal NO<sub>2</sub> variability. For model training, the targets are process-based NO<sub>2</sub> fields from the CAMS European air quality reanalyses (Inness et al., 2019; Peuch et al., 2022)

https://doi.org/10.5194/egusphere-2025-4259 Preprint. Discussion started: 24 November 2025

© Author(s) 2025. CC BY 4.0 License.

and real-world surface NO<sub>2</sub> measurements from the EEA AirBase network (European Environment Agency, 2024). CAMS supplies physically consistent large-scale 3D NO<sub>2</sub> distributions, while EEA data constrain the model to match local concentration patterns. Details on data preparation are provided in Section 2.2.


To effectively learn NO<sub>2</sub> patterns from diverse datasets, DACNO<sub>2</sub> employs an encoder–decoder architecture with five dedicated encoder branches, each tailored to a specific group of input features. The model structure is described in Section 2.3.

Model training is organized into three sequential phases. In Phase 1, a baseline model is pre-trained on process-based CAMS data. In Phase 2, the model is further trained with both process-based and measurement data, improving its ability to represent local NO<sub>2</sub> gradients. In Phase 3, the model is fine-tuned using recent measurements to reflect current NO<sub>2</sub> trends and support real-world applications. Details of the training approach are provided in Section 2.4.

## 2.2 Dataset Preparation

## 2.2.1 Input Features

DACNO<sub>2</sub> utilizes 38 input datasets, organized into five groups: temporal indicators, emission inventories and proxies, geography, single-level meteorology, and multi-level meteorology. Details of all input features and their sources are provided in Table 1.



120

The temporal indicator group consists of the day of the week and the daily number of flights. The day of the week captures regular human activity cycles, reflecting variability between weekdays and weekends. Data on the daily number of flights, aggregated for nine countries in the study area (Eurocontrol, 2025), can indicate irregular activity such as holiday periods or major events, which may help explain the irregular changes in NO<sub>2</sub> emissions. Emission inventories and proxies include anthropogenic NO<sub>x</sub> emission inventories, road density, population density, and nighttime light. These features provide direct and indirect measures of NOx emissions, with high-resolution proxies complementing inventories at finer spatial scales. All datasets are resampled to a 2 km grid using interpolation, averaging, or rasterization methods.

135 G ir w

Geographic datasets include land elevation and land cover, providing terrain context to the ML model. Elevation influences atmospheric transport by creating physical barriers that can trap pollutants (Giovannini et al., 2020), while land cover serves as a proxy for the location and type of surface emissions (Beelen et al., 2013). Land cover is categorized into five classes: artificial surfaces, agricultural areas, forests and semi-natural areas, wetlands, and water bodies, aggregated from the original 44 categories by the mode method (the most frequently occurring land cover type). Both elevation and land cover data are resampled to a 2 km grid.

140

145

Meteorological features provide atmospheric information from the surface through the free troposphere, obtained from the European Centre for Medium-Range Weather Forecasts (ECMWF) ERA5 hourly single-level and multilevel (pressure-level) datasets (Hersbach et al., 2020). We use 24-hour meteorological features for the target day. Meteorological data are horizontally resampled to a 16 km grid, for three reasons: (1) the native ERA5 resolution (0.25°, approximately 25 km) is coarser than 2 km, and bilinear interpolation would mainly introduce artificial smoothness rather than genuine fine-scale gradients; (2) retaining many meteorological variables at 2 km would impose a significant computational burden; and (3) the DACNO<sub>2</sub> architecture uses a hierarchical encoder-decoder, where upscaling and downscaling follow a factor-of-two scaling scheme (e.g., 2 km, 4 km, 8 km, 16 km). Although

155

160

the ERA5-Land data can provide higher resolution (0.1°, approximately 9 km), it only covers the continental areas, which is not consistent with the model application scope.

The day-of-week feature is normalized using sine and cosine transforms to retain its cyclical nature. Land cover is one-hot encoded to convert categorical data into a numerical format. All other input features are normalized with z-scores, based on the mean and standard deviation of the training set.

Notably, satellite-derived NO<sub>2</sub> products were deliberately excluded from the input features for two key reasons. First, frequent data gaps in satellite products, due to cloud cover and quality control, would propagate into the model's output, preventing the generation of continuous, gap-free fields. Second, this exclusion allows for an independent evaluation of the model against satellite observations and preserves the potential to use satellite data as an independent constraint in future work.

Table 1. Summary of inputs and training targets for the DACNO2 model development.

| Group                                  | Data                     | Spatial Resolution                 | Temporal<br>Resolution | Dimension   | Data Source                       |  |  |  |
|----------------------------------------|--------------------------|------------------------------------|------------------------|-------------|-----------------------------------|--|--|--|
|                                        |                          | Inj                                | puts                   |             |                                   |  |  |  |
| Temporal                               | Daily number of flights  | -                                  | Daily                  | 1D          | (Eurocontrol, 2025)               |  |  |  |
| indicator                              | Day of week              | -                                  | -                      | (time)      | -                                 |  |  |  |
|                                        |                          | Anthropogenic                      |                        |             |                                   |  |  |  |
|                                        | CAMS global emission     | sector                             | -                      |             |                                   |  |  |  |
|                                        | inventories              | $(0.1^{\circ} \times 0.1^{\circ})$ | (average for           |             | (Soulie et al., 2024)             |  |  |  |
|                                        | inventories              | Shipping sector                    | 2018)                  |             |                                   |  |  |  |
|                                        |                          | $(0.25^{\circ}\times0.25^{\circ})$ |                        |             |                                   |  |  |  |
| Emission                               | Road networks (five      |                                    |                        |             | Global Roads Inventory Projection |  |  |  |
| Emission<br>inventories and<br>proxies | types)                   | Vector data                        | -                      |             | (GRIP) global roads database      |  |  |  |
|                                        | types)                   |                                    |                        |             | (Meijer et al., 2018)             |  |  |  |
|                                        |                          |                                    | _                      |             | Annual global Visible Infrare     |  |  |  |
|                                        | Nighttime light          | 500 m                              | (average for           | 2D          | Imaging Radiometer Suite          |  |  |  |
|                                        |                          |                                    | 2019 to 2022)          | (latitude × | (VIIRS) dataset                   |  |  |  |
|                                        |                          | 1 km                               | ,                      | longitude)  | (Elvidge et al., 2021)            |  |  |  |
|                                        | Population               |                                    | _                      | iongitude)  | JRC-GEOSTAT 2018 gridde           |  |  |  |
|                                        | ropulation               | T KIII                             |                        | _           | dataset (Silva et al., 2021)      |  |  |  |
|                                        | Land elevation           | 90 m                               |                        |             | Multi-Error-Removed Improve       |  |  |  |
|                                        |                          |                                    | _                      |             | Terrain digital elevation mode    |  |  |  |
|                                        |                          | , v III                            |                        |             | (MERIT DEM) (Yamazaki et a        |  |  |  |
| Geography                              |                          |                                    |                        |             | 2017)                             |  |  |  |
| 87                                     | Land cover               |                                    |                        |             | Coordination of Information of    |  |  |  |
|                                        | (resampled to 5 classes  | 100m                               | _                      |             | the Environment (CORINE)          |  |  |  |
|                                        | via mode aggregation)    | 24422                              |                        |             | Land Cover 2018                   |  |  |  |
|                                        |                          |                                    |                        |             | (Feranec et al., 2016)            |  |  |  |
|                                        | Boundary layer height    |                                    |                        |             |                                   |  |  |  |
|                                        | Mean boundary layer      |                                    |                        |             | European Center for Medium        |  |  |  |
|                                        | dissipation              |                                    |                        | 3D          | Range Weather Forecasts           |  |  |  |
|                                        | Surface pressure         |                                    |                        | (latitude × | (ECMWF) ERA5 hourly time          |  |  |  |
| Meteorology                            | Dewpoint Temperature     | 0.25°                              | Hourly                 | longitude × | series data on single levels fro  |  |  |  |
|                                        | 2m Temperature           |                                    |                        | time)       | 1940 to the present (Hersbach     |  |  |  |
|                                        | 10 m U and V wind        |                                    |                        |             | al., 2020)                        |  |  |  |
|                                        | Leaf area index (for low |                                    |                        |             |                                   |  |  |  |
|                                        | and high vegetation)     |                                    |                        |             |                                   |  |  |  |

|                         | Mean evaporation rate  Mean snow rate  Mean surface net  radiation flux (short and long wave)  Mean total precipitation  rate  Geopotential |                                                | -                           | 4D                                              |                                                                    |
|-------------------------|---------------------------------------------------------------------------------------------------------------------------------------------|------------------------------------------------|-----------------------------|-------------------------------------------------|--------------------------------------------------------------------|
|                         | Vorticity Relative humidity Temperature U and V wind Vertical velocity                                                                      | 0.25°<br>8 layers from 1000<br>hPa to 550 hPa  |                             | (latitude ×<br>longitude ×<br>height ×<br>time) | ECMWF ERA5 hourly data on pressure levels from 1940 to the present |
|                         |                                                                                                                                             | Trainin                                        | g targets                   |                                                 |                                                                    |
| Process-based data      | CAMS European air quality reanalysis NO <sub>2</sub>                                                                                        | 0.1°<br>8 layers from the<br>surface to 5000 m | Daily (average from hourly) | 3D<br>(latitude ×<br>longitude ×<br>height)     | (Peuch et al., 2022; Inness et al., 2019)                          |
| Surface<br>measurements | EEA AirBase surface<br>NO <sub>2</sub>                                                                                                      | -                                              | Hourly                      | 2D (latitude × longitude)                       | EEA AirBase network (European Environment Agency, 2024)            |

## 2.2.2 Training Targets




The training targets include CAMS European air quality reanalysis profile data (CAMS NO<sub>2</sub>) and in-situ measurements from the EEA AirBase network (EEA NO<sub>2</sub>). The datasets are both listed in Table 1. CAMS NO<sub>2</sub> offers extensive and continuous 3D NO<sub>2</sub> data aligned with physical and chemical processes, while EEA NO<sub>2</sub> provides ground-based in-situ measurements from sparsely distributed monitoring stations.

CAMS  $NO_2$  is the median ensemble of 11 different regional models (Inness et al., 2019; Peuch et al., 2022). The dataset provides hourly  $NO_2$  distributions at eight vertical heights above the surface (surface, 50m, 100m, 250m, 500m, 750m, 1000m, 2000m, 3000m, and 5000m) and has a horizontal resolution of  $0.1^{\circ} \times 0.1^{\circ}$  (10 km  $\times$  10 km). CAMS  $NO_2$  has assimilated EEA observations and includes both interim and validated reanalyses. Interim data relies on near-real-time observations without full validation, whereas validated data undergo rigorous quality control with an additional delay. In this study, we used CAMS  $NO_2$  data from 2019 to 2023, where the 2019–2021 data are validated reanalysis data and the 2022–2023 data are interim reanalysis data, based on data availability.






CAMS NO<sub>2</sub> was processed by averaging hourly data to daily values and by bilinearly interpolating its horizontal resolution from 10 km to 8 km to match the model's scaling scheme. In addition, CAMS NO<sub>2</sub> concentrations at each vertical level were rescaled by multiplying them by the inverse of the ratio between the mean NO<sub>2</sub> concentration at that level and the mean surface-level NO<sub>2</sub> concentration, where the ratio was calculated from the training dataset. This adjustment ensures that the model gives adequate attention to higher-altitude NO<sub>2</sub> concentrations, which are otherwise much lower than surface values and could be neglected during training (Li and Xing, 2024; Kuhn et al., 2024a). During model inference, the predicted NO<sub>2</sub> concentrations at each level were multiplied by the corresponding ratio to restore the original vertical profile.

EEA NO<sub>2</sub> was collected from background and industrial monitoring stations (European Environment Agency, 2024) and mapped onto a 2 km grid. Traffic stations were excluded because their measurements represent a very local area, significantly smaller than the 2 km grid cells of our study. If multiple stations were located within the same grid cell, their values were averaged. When both background and industrial stations existed in a grid cell, the cell was classified as background. In total, 773 grid cells with measurements were identified, with 575 assigned for training and 198 for final evaluation. Because background EEA NO<sub>2</sub> is assimilated into CAMS, the split of background stations followed the CAMS model assimilation system (Copernicus Atmosphere Monitoring Service, 2024) to prevent data leakage, while industrial stations were randomly split. All EEA NO<sub>2</sub> data were converted from hourly to daily averages.

## 2.2.3 Patch-Based Data Processing and Reconstruction

To balance the model's receptive field and computational efficiency, we used a patching method. Specifically, all datasets except the temporal indicator were divided into patches of 512 km × 512 km with partial overlap. This produced grid sizes of 32 × 32 for ERA5 meteorological data, 64 × 64 for CAMS NO<sub>2</sub> data, and 256 × 256 for emission inventories and proxies, geographic data, and EEA NO<sub>2</sub> data. In this study, each patch was treated as a single input sample for the model, and 12 samples were generated for one day. More samples can be generated as needed by reducing the stride of the sliding window. During model inference, the output patches were merged using a weighted averaging scheme based on a 2D Hann window (Oppenheim, 1999), which assigns lower weights to patch edges and higher weights to central regions. For each grid cell, weighted values from all overlapping patches were summed and normalized by the total weights. This reconstruction method reduced edge artifacts in overlapping areas and ensured smooth transitions across patch boundaries.


#### 2.3 Model Architecture and Design

Figure 2. DACNO<sub>2</sub> model architecture. The model features a multi-branch encoder-decoder design for daily 3D NO<sub>2</sub> prediction. Five input groups are processed separately: ERA5 single-level meteorological variables (ConvLSTM-2D), ERA5 multi-level meteorological variables (ConvLSTM-3D), emission inventories and proxies (2D CNN), geography (2D CNN), and temporal features (MLP-based embedding fusion). Outputs from all encoder branches are fused and passed into a unified 3D CNN decoder to generate high-resolution NO<sub>2</sub> fields. The architecture enables the extraction of spatial, temporal, and multi-level atmospheric features, supporting fine-scale NO<sub>2</sub> modeling. Input and output dimensions are indicated for each module.

The architecture of the DACNO<sub>2</sub> model is illustrated in Fig. 2. The model adopts an encoder–decoder framework with residual connections (He et al., 2016) to map multi-source input features to the daily 3D NO<sub>2</sub> field. DACNO<sub>2</sub> integrates several types of neural network modules, including multilayer perceptron (MLP), convolutional neural network (CNN), and convolutional long short-term memory (ConvLSTM), to process and fuse heterogeneous input tensors. Each module is chosen for its specific strengths in handling different data structures. ConvLSTM is for spatiotemporal sequences, CNN is for spatial hierarchies, and MLP is for tabular feature vectors. Inception-style structures are applied in several neural network modules to enable the model to capture both local-scale and broader-scale spatial features.







#### 2.3.1 Encoder and Decoder

DACNO2 encodes ERA5 single-level (hourly 2D) and multi-level (hourly 3D) meteorological data using ConvLSTM-2D and ConvLSTM-3D modules, respectively. Both modules are based on the ConvLSTM architecture proposed by Shi et al. (2015), which combines convolutional layers for spatial feature extraction with long short-term memory (LSTM) units for temporal sequence modeling. ERA5 data are processed using a progressive upscaling strategy, where the horizontal grid size increases stepwise from  $32 \times 32$  to  $64 \times 64$ ,  $128 \times 128$ , and  $256 \times 256$ , while the vertical dimension remains at 8 for multi-level inputs. This upscaling preserves spatial detail and enables residual connections to the decoder, unlike conventional encoders that downsample feature maps. To manage computational cost, the temporal sequence length is halved after each ConvLSTM block through subsampling, resulting in sequence lengths of 24, 12, 6, and 3 at successive stages. At each stage, the last time slice is extracted for feature fusion.

Emission and geographic variables are encoded by dedicated 2D CNN blocks, which extract hierarchical spatial features as the resolution decreases from  $256 \times 256$  to  $32 \times 32$ . At the  $32 \times 32$  latent stage, features from all four branches are passed through CNN-based transition layers, each forming a 3D tensor. For each branch, feature values are assigned only to physically relevant vertical levels within the tensor, while all other levels are set to zero. Specifically, emission and geographic features are assigned to the surface level, ERA5 single-level features are placed in the lowest five levels, and ERA5 multi-level features span all vertical levels. The resulting tensors are concatenated along the channel dimension and fused using a 3D CNN block. Temporal indicators are encoded by an MLP, then expanded to match the latent spatial dimensions, and integrated at this stage, allowing the model to capture both spatial and temporal context. The same feature fusion scheme is applied to residual connections between the encoder and decoder at multiple spatial scales, although temporal embedding is used only at the  $32 \times 32$  stage.

The decoder uses 3D CNN modules with hierarchical upscaling from  $32 \times 32$  to  $256 \times 256$  in the horizontal dimension, while maintaining a vertical size of 8. This structure learns spatial correlations across multiple altitude levels and captures both horizontal and vertical dependencies in NO<sub>2</sub> distributions. All 2D and 3D CNN blocks use the sigmoid linear unit (SiLU) activation function (Elfwing et al., 2017), while the output layer uses the softplus activation function to ensure non-negative estimates of the 3D NO<sub>2</sub> field.




## 2.3.2 Inception-Based Modules

To enhance multi-scale feature extraction, DACNO<sub>2</sub> incorporates inception modules throughout its architecture (Fig. 2), inspired by the work of inception architecture (Szegedy et al., 2014; Szegedy et al., 2015). The core concept of this architecture is to use parallel convolutional operations with varying kernel sizes, enabling the model to efficiently capture both fine-scale and broad features simultaneously. In the ConvLSTM-2D and ConvLSTM-3D branches, each inception block applies parallel convolutional operations with varying kernel sizes ( $1\times1$ ,  $3\times3$ ,  $5\times5$ ) and a max-pooling branch, enabling the model to capture both local and broader spatiotemporal patterns. The 2D CNN modules extend this approach, combining parallel  $1\times1$ ,  $3\times3$ , and  $5\times5$  convolutions, a factorized  $7\times7$  path (decomposed into  $1\times7$  and  $7\times1$  convolutions), and a pooling branch. For 3D CNN modules, inception blocks use parallel convolutions with different spatial and vertical kernel shapes, such as  $1\times1\times1$ ,  $1\times3\times3$ , and  $3\times1\times1$ , along with a pooling branch. In all cases, each parallel branch includes its own batch normalization, activation, and dropout, after which the outputs are concatenated along the channel dimension. A similar design has been applied in a previous deep learning model for NO<sub>2</sub> estimation (Zhang et al., 2022a). It enables the model to effectively integrate information across multiple spatial and vertical scales, improving the representation of complex atmospheric NO<sub>2</sub> distributions.




## 2.4 Three-Phase Training Strategy

The DACNO<sub>2</sub> model development employs a three-phase training strategy, including pre-training, multi-constraint fine-tuning, and adaptive fine-tuning. Such a strategy enables the model to learn general patterns (e.g., a-priori knowledge) from a broad dataset and then transfer this internal knowledge to improve its performance on a new, more specific task. Similar approaches have been widely adopted in the development of artificial intelligence (AI) models across various domains such as earth system modeling, large language models, and biomedical image analysis (Zhuang et al., 2019; Zhou et al., 2017; Ding et al., 2023; Bodnar et al., 2024).

#### 2.4.1 Phase-1

In the first phase, the DACNO<sub>2</sub> model was pre-trained on the CAMS NO<sub>2</sub> data. This dataset provides physically consistent 3D NO<sub>2</sub> distributions by assimilating real-world observations into chemical transport models (Inness et al., 2019), enabling the model to learn comprehensive 3D NO<sub>2</sub> patterns governed by broad-scale atmospheric processes. This approach is inspired by recent progress in AI weather modeling (Bi et al., 2023; Lam et al., 2023) and the earth system foundation model (Bodnar et al., 2025), which uses ERA5 and CAMS data for 3D forecasting of weather and air quality.

In this step, the training loss is defined as the sum of the Mean Squared Error (MSE) loss and the Structural Similarity Index Measure (SSIM) loss (Zhao et al., 2017; Zhou et al., 2004) between the DACNO<sub>2</sub> prediction and the CAMS NO<sub>2</sub> data at 8 km resolution.

$$Loss = LossMSE_{DACNO_2-CAMS} + LossSSIM_{DACNO_2-CAMS}$$
 (1)

MSE quantifies the absolute differences in NO<sub>2</sub> concentrations, while SSIM evaluates the similarity of spatial patterns between model outputs and the CAMS reference. SSIM is computed independently at each vertical level by comparing normalized 2D horizontal slices of the predicted and reference NO<sub>2</sub> fields. Specifically, each slice is min-max normalized to the range of 0 to 1 prior to SSIM calculation, ensuring that the SSIM loss reflects only structural similarity rather than magnitude differences. The final SSIM loss is calculated as one minus the mean SSIM across all vertical levels. This dual-loss formulation encourages the model to match both the overall concentration values and the detailed spatial structures of 3D NO<sub>2</sub> fields.

The model was trained and validated using a random sample split from the 2019, 2021, and 2022 datasets (80% for training, 20% for validation), with 2023 reserved as an independent test set. Data from 2020 was excluded





from this process because a preliminary experiment showed that its inclusion substantially degraded the model performance on the unknown period (i.e., 2022 data, which was initially held out as an independent validation year in that experiment). This might be due to the unexpectedly higher NO<sub>2</sub> concentrations above 1000 m in that year (Fig. S1), which is also documented in the CAMS Evaluation and Quality Control (EQC) report (Meleux et al., 2023). While the cause remains unclear, we speculate that this anomaly is related to the substantial decrease in NO<sub>x</sub> emissions during 2020 due to the COVID-19 pandemic (Levelt et al., 2022) and not well accounted for in the CAMS model. We evaluate and discuss DACNO<sub>2</sub> performance for that special year in Section 4.4.

## 2.4.2 Phase-2

In the second phase, we fine-tuned DACNO<sub>2</sub>-Phase-1 by introducing an additional MSE constraint based on EEA NO<sub>2</sub>, while maintaining the CAMS NO<sub>2</sub> constraints, as shown in Equation (2). The EEA NO<sub>2</sub> MSE was computed at the surface level and only for 2 km grids with available EEA data

$$Loss = Loss MSE_{DACNO_2-CAMS} + Loss SSIM_{DACNO_2-CAMS} + Loss MSE_{DACNO_2-EEA}$$
 (2)

The EEA NO<sub>2</sub> data were split into training and validation sets using the same spatiotemporal alignment as the CAMS NO<sub>2</sub> split. Most training settings remained consistent with the first phase, except that the learning rate was reduced and the EEA NO<sub>2</sub> MSE term was added to both the training loss and the validation metric. The model with the best validation performance was selected and is referred to as DACNO<sub>2</sub>-Phase-2 for subsequent use. Although Phase-2 includes the same CAMS constraint as Phase-1, which may make Phase-1 appear redundant, we recommend retaining Phase-1. Skipping directly to Phase-2 can cause the model to overfit local EEA observations and limit its ability to learn generalizable NO<sub>2</sub> patterns from process-based data.

#### 315 **2.4.3 Phase-3**

Recent changes in air quality policies and emission technologies (Castellanos and Boersma, 2012; Wang et al., 2021; Chang et al., 2023) may introduce systematic NO<sub>2</sub> variations that are not well represented in the historical training dataset. To ensure the DACNO<sub>2</sub> model remains adaptable to such real-world changes, we introduced a third phase. In this step, we adopted a strategy inspired by the data assimilation system in the CAMS model (Inness et al., 2019). DACNO<sub>2</sub>-Phase-2 was fine-tuned using only EEA NO<sub>2</sub> data from training stations during the test period (2023 in this study), to mimic real-world application. To maintain spatial patterns learned from earlier phases, a regularization term based on SSIM was added to both the training loss and validation metric. SSIM was computed at 8 km resolution between predictions from the updated model and DACNO<sub>2</sub>-Phase-2 (equation 3):

$$Loss = LossMSE_{DACNO_2(new)-EEA} + LossSSIM_{DACNO_2(new)-DACNO_2(Phase-2)}$$
(3)

https://doi.org/10.5194/egusphere-2025-4259 Preprint. Discussion started: 24 November 2025



325 This approach allows the updated model to adjust prediction magnitudes in response to new measurements while preserving spatial patterns established in previous phases, since the CAMS constraint is no longer available in Phase-3. The model with the best validation performance was selected and is referred to as DACNO<sub>2</sub>-Phase-3, which incorporates recent real-world NO<sub>2</sub> variations while retaining consistency with patterns learned during earlier training.

## 2.4.4 Training and Implementation

DACNO<sub>2</sub> was trained and implemented in Python using PyTorch on two NVIDIA A30 GPUs. Training was performed with a batch size of 56, achieved by gradient accumulation. The first and second training phases each required approximately three weeks to complete 200 epochs on three years of data. The third training phase required about one week for 100 epochs on a single year of data. Once trained, the model generates daily NO2 estimates for the whole area within minutes. Further efficiency improvements are possible through hardware upgrades or model optimization.

## 3. Assessing DACNO<sub>2</sub> Performance and Evolution

# 3.1 Model Performance Across Training Phases

Table 2. Performance of DACNO2 on the 2023 test dataset.

|              | DAG                      | DACNO <sub>2</sub> -Phase-1 DACNO <sub>2</sub> -Phase-2 |                |       | DACNO <sub>2</sub> -Phase-3 |      |                |       | CAMS-2km     |      |                |       |              |      |                |              |  |
|--------------|--------------------------|---------------------------------------------------------|----------------|-------|-----------------------------|------|----------------|-------|--------------|------|----------------|-------|--------------|------|----------------|--------------|--|
| EEA          | RMSE (ug/m³)             | r                                                       | $\mathbb{R}^2$ |       | RMSE (ug/m³)                | r    | $\mathbb{R}^2$ |       | RMSE (ug/m³) | r    | R <sup>2</sup> |       | RMSE (ug/m³) | r    | $\mathbb{R}^2$ | bias (ug/m³) |  |
| Total        | 5.88                     | 0.75                                                    | 0.52           | -0.93 | 5.81                        | 0.79 | 0.54           | 1.67  | 4.99         | 0.82 | 0.66           | -0.38 | 5.32         | 0.80 | 0.61           | -1.15        |  |
| Urban        | 6.45                     | 0.76                                                    | 0.47           | -2.57 | 5.64                        | 0.80 | 0.59           | 0.49  | 5.34         | 0.82 | 0.64           | -1.42 | 5.89         | 0.82 | 0.56           | -2.89        |  |
| Suburban     | 5.11                     | 0.79                                                    | 0.58           | -0.04 | 6.01                        | 0.81 | 0.42           | 2.87  | 4.55         | 0.83 | 0.67           | 0.58  | 4.39         | 0.84 | 0.69           | -0.23        |  |
| Rural        | 5.22                     | 0.76                                                    | 0.42           | 1.96  | 5.96                        | 0.76 | 0.25           | 3.17  | 4.59         | 0.78 | 0.55           | 1.05  | 4.79         | 0.80 | 0.51           | 1.98         |  |
|              | DACNO <sub>2</sub> -10km |                                                         |                |       |                             |      |                |       |              |      |                |       |              |      |                |              |  |
| Total levels | 0.99                     | 0.96                                                    | 0.91           | -0.05 | 1.17                        | 0.96 | 0.87           | 0.25  | 1.02         | 0.95 | 0.90           | -0.10 |              |      |                |              |  |
| L0           | 1.92                     | 0.94                                                    | 0.85           | -0.36 | 2.29                        | 0.93 | 0.79           | 0.75  | 1.94         | 0.93 | 0.85           | -0.63 |              |      |                |              |  |
| L50          | 1.58                     | 0.95                                                    | 0.88           | -0.25 | 1.94                        | 0.94 | 0.82           | 0.66  | 1.62         | 0.94 | 0.87           | -0.31 |              |      |                |              |  |
| L250         | 1.05                     | 0.93                                                    | 0.83           | 0.10  | 1.19                        | 0.93 | 0.78           | 0.41  | 1.09         | 0.93 | 0.82           | 0.01  | CAMS (10km)  |      |                |              |  |
| L500         | 0.70                     | 0.92                                                    | 0.79           | 0.09  | 0.71                        | 0.92 | 0.79           | 0.16  | 0.79         | 0.90 | 0.74           | 0.05  |              |      |                |              |  |
| L1000        | 0.31                     | 0.89                                                    | 0.72           | 0.03  | 0.30                        | 0.89 | 0.74           | 0.03  | 0.39         | 0.86 | 0.56           | 0.06  |              |      |                |              |  |
| L2000        | 0.08                     | 0.81                                                    | 0.61           | -0.01 | 0.08                        | 0.81 | 0.64           | -0.01 | 0.09         | 0.79 | 0.48           | 0.00  |              |      |                |              |  |





| L3000 | 0.04 | 0.78 0.55 -0.0 | 0.03 | 0.77 0.58 0.0 | 0.04   | 0.74 0.40  | 0.00 |
|-------|------|----------------|------|---------------|--------|------------|------|
| L5000 | 0.01 | 0.67 0.31 0.00 | 0.01 | 0.66 0.34 0.0 | 0 0.02 | 0.61 -0.01 | 0.00 |

Note: For the comparison against EEA NO<sub>2</sub> (shown in the upper panel), both DACNO<sub>2</sub> outputs and CAMS NO<sub>2</sub> were evaluated at 2 km grids. The CAMS NO<sub>2</sub> data was interpolated to a 2 km resolution (CAMS-2km) and used as a baseline in this comparison. For evaluating DACNO<sub>2</sub> using CAMS NO<sub>2</sub> (shown in the lower panel), DACNO<sub>2</sub> outputs were downsampled and evaluated at the original 10 km resolution of CAMS across all vertical levels as well as for individual levels.

The performance of the DACNO<sub>2</sub> model was evaluated using both EEA NO<sub>2</sub> and CAMS NO<sub>2</sub> test data from 2023. For the comparison against EEA NO<sub>2</sub> (results in the upper panel of Table 2), both DACNO<sub>2</sub> outputs and CAMS NO<sub>2</sub> were evaluated at 2 km grids. The CAMS NO<sub>2</sub> data was interpolated to 2 km resolution (CAMS-2km) and served as a baseline in this comparison. For the evaluation of DACNO<sub>2</sub> using CAMS NO<sub>2</sub> (results in the lower panel of Table 2), DACNO<sub>2</sub> outputs were evaluated at the CAMS original 10 km resolution across all vertical levels as well as for individual levels. Evaluation metrics included root mean squared error (RMSE), Pearson correlation coefficient (r), coefficient of determination (R<sup>2</sup>), and bias.

Comparisons with EEA NO<sub>2</sub> indicate progressive improvement across DACNO<sub>2</sub> training phases. DACNO<sub>2</sub>-Phase-3 achieves the best overall agreement (RMSE = 4.99 ug/m³, r = 0.82,  $R^2 = 0.66$ , bias = -0.38 ug/m³), outperforming both DACNO<sub>2</sub>-Phase-1 (RMSE = 5.88 ug/m³, r = 0.75,  $R^2 = 0.52$ , bias = -0.93 ug/m³) and DACNO<sub>2</sub>-Phase-2 (RMSE = 5.81 ug/m³, r = 0.79,  $R^2 = 0.54$ , bias = -1.67 ug/m³). This demonstrates the added value of incorporating local EEA constraints and adaptive fine-tuning. Compared to the interpolated CAMS-2km dataset (RMSE = 5.32 ug/m³, r = 0.80,  $R^2 = 0.61$ , bias = -1.15 ug/m³), DACNO<sub>2</sub>-Phase-3 shows improved accuracy and reduced bias. Station-type analysis further highlights the advantages of DACNO<sub>2</sub>-Phase-3, especially at urban and rural sites. For urban locations, DACNO<sub>2</sub>-Phase-3 achieves better agreement (RMSE = 5.34 ug/m³, r = 0.82,  $R^2 = 0.64$ , bias = -1.42 ug/m³) compared with CAMS-2km (RMSE = 5.89 ug/m³, r = 0.82,  $R^2 = 0.56$ , bias = -2.89 ug/m³). In rural areas, DACNO<sub>2</sub>-Phase-3 reduces the bias (RMSE = 4.59 ug/m³, bias = 1.05 ug/m³) compared to CAMS-2km (RMSE = 4.79 ug/m³, bias = 1.98 ug/m³). These results suggest that DACNO<sub>2</sub> is more effective than CAMS at capturing localized emission patterns and small-scale spatial variability in areas with either dense emission sources or diffuse background levels.



Comparisons with CAMS NO<sub>2</sub> across all levels show that DACNO<sub>2</sub> effectively learns and preserves 3D NO<sub>2</sub> distributions across all training phases. Near the surface (Level 0 m), DACNO<sub>2</sub>-Phase-3 maintains strong agreement with CAMS (RMSE = 1.94  $\mu$ g/m³, r = 0.93, R² = 0.85, bias = -0.63  $\mu$ g/m³), and performance remains robust at mid-altitudes (Level 500 m: RMSE = 0.79  $\mu$ g/m³, r = 0.90, R² = 0.74, bias = 0.05  $\mu$ g/m³), similar to earlier phases. At higher levels, particularly 5000 m, the agreement decreases, with DACNO<sub>2</sub>-Phase-3 yielding a near-zero R² (-0.01) compared to 0.31 in DACNO<sub>2</sub>-Phase-1 and 0.34 in DACNO<sub>2</sub>-Phase-2. However, Pearson correlation remains moderate (r  $\geq$  0.6), indicating that differences at these heights are mainly due to magnitude adjustment rather than loss of spatial structure. The very low NO<sub>2</sub> concentrations (approximately 0.1  $\mu$ g/m³ at 3000 m and 0.03  $\mu$ g/m³ at 5000 m; see Fig. S1) at these high levels also increase relative noise and contribute to metric variability. This reduction in agreement at upper levels mainly reflects the greater uncertainty and lack of direct constraints at high altitudes, which remains a key challenge for ML-based 3D air quality modeling.



#### 3.2 Model Evolution in the Multi-constraint Strategy

Figure 3. Spatial comparison of surface  $NO_2$  estimates for 2023 from multiple models. (a) Annual mean surface  $NO_2$  fields over the entire study region from DACNO<sub>2</sub>-Phase-1, DACNO<sub>2</sub>-Phase-2, DACNO<sub>2</sub>-Phase-3, CAMS (10 km), and CAMS-2km (bilinearly interpolated to 2 km). (b-d) Enlarged views for three representative local areas: (b) Paris, (c) the northern region ( $NO_2$  hotspot area encompassing the Netherlands, Belgium, and the Ruhr area), and (d) the Alpine region.

To further illustrate the evolution of estimated NO<sub>2</sub> spatial distributions achieved through a phased training, multiconstraint strategy, Fig. 3 compares average surface NO<sub>2</sub> estimates for 2023 from DACNO<sub>2</sub>-Phase-1, DACNO<sub>2</sub>-Phase-2, DACNO<sub>2</sub>-Phase-3, CAMS, and CAMS-2km. Results are shown for the full study region and three representative local areas of Paris, the northern region (NO<sub>2</sub> hotspot area encompassing the Netherlands, Belgium, and the Ruhr area), and the Alpine region.

Across the study region (Fig. 3a), all models exhibit broad and similar NO<sub>2</sub> patterns over land and ocean, consistent with the high spatial agreement between DACNO<sub>2</sub> and CAMS NO<sub>2</sub> reported in Section 3.1.




Nonetheless, DACNO<sub>2</sub>-Phase-2 and DACNO<sub>2</sub>-Phase-3 yield sharper spatial contrasts and more clearly defined local NO<sub>2</sub> hotspots than CAMS and DACNO<sub>2</sub>-Phase-1. As an additional experiment, we trained the model using only EEA NO<sub>2</sub> data, resulting in the DACNO<sub>2</sub>-onlyobs version. As shown in Fig. S2, this model yields effective NO<sub>2</sub> estimates primarily limited to the land surface and cannot reproduce the shipping track patterns, which are visible in the CAMS and DACNO<sub>2</sub> results. Meanwhile, this model produces obvious artifacts over the ocean and at higher altitudes due to the lack of training constraints. These differences highlight the significance of the CAMS NO<sub>2</sub> constraint in facilitating broad spatial generalization in ML-based models.

Differences between models become more pronounced when focusing on local regions (Figs. 3b–d). CAMS NO<sub>2</sub> exhibits visible pixelation effects in these areas due to its coarse native resolution. While bilinear interpolation (as in CAMS-2km) can smooth these effects, it does not introduce additional spatial detail, resulting in oversmoothed patterns. DACNO<sub>2</sub>-Phase-1 shows a spatial NO<sub>2</sub> distribution similar to CAMS-2km, despite using high-resolution input features from emission proxies and geography. This indicates that constraints from CAMS NO<sub>2</sub> alone are insufficient for the model to capture fine-scale local NO<sub>2</sub> variability. Incorporating the EEA NO<sub>2</sub> constraint in DACNO<sub>2</sub>-Phase-2 addresses this limitation, inspired by approaches in recent ML-based high-resolution surface NO<sub>2</sub> modeling studies using ground measurements as targets (Sun et al., 2024; Wei et al., 2022; Kim et al., 2021; Ghahremanloo et al., 2023). DACNO<sub>2</sub>-Phase-2 reconstructs spatial patterns of NO<sub>2</sub> that better match urban layout in Paris (Fig. 3b), identifies more small-scale emission hotspots in the northern region (Fig. 3c), and enhances hotspot signals in the Alpine region (Fig. 3d). DACNO<sub>2</sub>-Phase-3 retains these spatial characteristics and primarily adjusts concentration magnitudes by assimilating new measurements to better represent actual NO<sub>2</sub> levels during the application period. For example, the average surface NO<sub>2</sub> concentration estimate in Paris decreases from 11.89 μg/m³ in DACNO<sub>2</sub>-Phase-2 to 10.08 μg/m³ in DACNO<sub>2</sub>-Phase-3. This evolution demonstrates the value of integrating multiple constraints and adaptive fine-tuning for high-resolution NO<sub>2</sub> estimation.


## 3.3 Global and Local Differences Between DACNO2 and CAMS

420 Figure 4. Annual mean NO<sub>2</sub> distributions for 2023 estimated from DACNO<sub>2</sub>-Phase-3 (2 km) and CAMS (10 km) at multiple vertical levels. Level-wise average NO<sub>2</sub> distributions over (a) Western Europe (entire study region), (b) Paris, (c) the northern region, and (d) the Alpine region.

To further analyze differences in 3D NO<sub>2</sub> estimates between DACNO<sub>2</sub> and CAMS, Fig. 4 compares their annual average NO<sub>2</sub> distributions for 2023 across all vertical levels over the entire study region and three selected local areas. At the regional scale (Fig. 4a), DACNO<sub>2</sub> and CAMS show strong overall agreement at all altitudes, demonstrating that DACNO<sub>2</sub> effectively learns and reproduces large-scale 3D NO<sub>2</sub> structures from CAMS. However, DACNO<sub>2</sub> provides enhanced spatial detail, presenting sharper gradients and better-defined urban and industrial hotspots, particularly from the surface up to 250 m. At higher altitudes, the differences between the two models gradually diminish, accompanied by a decrease in NO<sub>2</sub> concentrations. Nevertheless, subtle magnitude discrepancies persist, with DACNO<sub>2</sub> estimates reaching lower values, sometimes approaching zero.

Local-scale comparisons further highlight the advantages of DACNO<sub>2</sub> (Figs. 4b–d). In the Paris region (Fig. 4b), DACNO<sub>2</sub> provides finer spatial detail and stronger NO<sub>2</sub> signals at lower altitudes (e.g., 0 m: 10.08 µg/m<sup>3</sup>; 50 m:

8.94 μg/m³; 250 m: 4.65 μg/m³), whereas CAMS results remain coarser with generally lower estimates (0 m: 8.43 μg/m³; 50 m: 7.15 μg/m³; 250 m: 3.63 μg/m³). In the northern region (Fig. 4c), DACNO<sub>2</sub> more distinctly resolves localized emission sources at low levels, capturing a greater number of hotspots than CAMS. As a result, the average NO<sub>2</sub> concentration from DACNO<sub>2</sub> is elevated throughout the boundary layer (up to 1000 m), with mean values 8.8% higher than those from CAMS. In the Alpine region (Fig. 4d), DACNO<sub>2</sub> more effectively represents terrain-driven gradients and captures NO<sub>2</sub> signals within mountainous areas, demonstrating greater sensitivity to complex topographic influences. At higher altitudes, fine-scale variability diminishes in both models and their predicted NO<sub>2</sub> fields become more similar. This is because the influence of local emissions and surface features weakens, while regional-scale processes and long-range transport dominate (see Section 4.1). This reduced difference is also accompanied by much lower NO<sub>2</sub> concentrations at higher altitudes.

## 4. Insights and Implications of DACNO<sub>2</sub>

## 4.1 Feature Importance and Data-driven Insights

Figure 5. Relative importance of each input feature group for DACNO2 model predictions, evaluated using the integrated gradients (IG) method. (a) Feature group contributions to RMSE between DACNO2 surface NO2 estimates and EEA ground-based measurements for 2023. (b) Feature group contributions to RMSE between DACNO2 and CAMS NO2 estimates at different vertical levels for 2023. The five feature groups are: temporal indicators, emission inventories and proxies, geography, ERA5 single-level meteorology, and ERA5 multi-level meteorology. Results are shown for each model training phase (Phase-1, Phase-2, and Phase-3), illustrating how the relative influence of input feature groups varies with training constraints and altitude. See Figure S3 for the contributions of individual features within each group.

We assessed the relative importance of input feature groups in DACNO<sub>2</sub> using the integrated gradients (IG) method (Sundararajan et al., 2017) implemented via the Captum interpretability library (Kokhlikyan et al., 2020). IG quantifies the effect of varying each input feature from a zero baseline to its actual value on a selected target function. In this analysis, we computed IG at two targets: (1) the RMSE between DACNO<sub>2</sub> predictions and 2023 EEA NO<sub>2</sub> training measurements at the surface, and (2) the RMSE between DACNO<sub>2</sub> predictions and 2023 CAMS






NO<sub>2</sub> at multiple vertical levels. Feature group results are shown in Fig. 5, and results for individual features are provided in Fig. S3.

For surface NO<sub>2</sub> predictions evaluated against EEA measurements, DACNO<sub>2</sub> relies primarily on emission proxies, geographic features, and multi-level meteorological variables, while temporal indicators and single-level meteorological features play a lesser role. The addition of the EEA NO<sub>2</sub> constraint in Phase-2 and Phase-3 increases the importance of geographic data, highlighting its value for high-resolution surface NO<sub>2</sub> estimation. As shown in Fig. S3, land cover emerges as the most influential single feature in DACNO<sub>2</sub>-Phase-3, accounting for 36.6% of the total importance. Multi-level meteorological variables dominate the meteorological contribution, although the radiation variable from the single-level meteorology group also remains relevant (1.0% in Phase-3, Fig. S3). This likely reflects some redundancy between single-level and multi-level meteorological features, with multi-level data providing more comprehensive atmospheric information.

For NO<sub>2</sub> estimates by level evaluated against CAMS, the distribution of input feature importance at lower levels (up to 1000 m) is similar to that for surface NO<sub>2</sub> evaluated against EEA, suggesting that DACNO<sub>2</sub> remains relatively stable across training phases with different constraints. Differences between the three-phase models are most apparent near the surface but gradually diminish with height. The importance of geographic features steadily decreases as height increases, while emission features reach their highest influence around 500 m before declining. Above 3000 m, both become negligible, reflecting the transition from the Planetary Boundary Layer (PBL), which is influenced by local surface features, into the free troposphere, which is dominated by broad-scale processes.

In contrast, temporal indicators, single-level meteorology, and especially multi-level meteorological features become increasingly important with height. For example, in DACNO<sub>2</sub>-Phase-3, the contribution from multi-level meteorology rises from 16.5% at the surface to 71.8% at 5000 m, while temporal indicators and single-level meteorology also show steady increases. This shift highlights the greater reliance on temporal and large-scale atmospheric information for NO<sub>2</sub> estimates at higher levels. Among these features, radiation flux (3.3% at 3000 m, 2.4% at 5000 m) primarily drives the increase in single-level meteorology, while wind becomes the dominant variable among all meteorological features (approximately 21% at both 3000 m and 5000 m, Fig. S3). Given the consistently low overall contribution of single-level meteorological variables, future model development may consider reducing or refining the use of this feature group to streamline the input space.

## 4.2 Enhanced Vertical NO<sub>2</sub> Profile Representation

Figure 6. Comparison of NO<sub>2</sub> profile estimates from DACNO<sub>2</sub> (2 km resolution) and CAMS (10 km resolution) for the Paris and Alpine regions in 2023. Regional average vertical profiles and surface NO<sub>2</sub> distributions for Paris (a) and the Alpine region (b), with results shown over Google Earth imagery. Interpixel variability of NO<sub>2</sub> profiles from DACNO<sub>2</sub> and CAMS along a 100 km transect (black boxes) in the Paris area (c) and a 200 km transect in the Alpine region (d), illustrating local-scale differences in vertical structure.

Fig. 6 compares the average 2023 NO<sub>2</sub> profile estimates from DACNO<sub>2</sub> (2 km) and CAMS (10 km) for the Paris and Alpine regions, with results overlaid on Google Earth imagery for geographic context. In Paris, the regional




average profile (Fig. 6a) indicates that DACNO<sub>2</sub> yields higher near-surface NO<sub>2</sub> concentrations and steeper vertical gradients compared to CAMS. This enhancement likely results from DACNO<sub>2</sub>'s use of high-resolution emission proxies and land cover information, allowing the model to resolve smaller and more localized emission sources (Kuik et al., 2018; Shahrokhishahraki et al., 2022). At the local scale, we take transect over the grids of 100 km over Paris to compare the interpixel profile variability from CAMS (10 km) and DACNO<sub>2</sub> (2 km) (Fig. 6c). It is observed that DACNO<sub>2</sub> more clearly distinguishes spatial variability in the vertical structure, showing sharper contrasts and more pronounced local peaks than CAMS, particularly below 250 m. The regional average profiles for the Alpine area are similar between DACNO<sub>2</sub> and CAMS (Fig. 6b), which is due to the overall lower concentrations over this region with limited emission sources. However, local differences remain visible across a 200 km transect (Fig. 6d). DACNO<sub>2</sub> captures higher NO<sub>2</sub> concentrations around urban and small-scale hotspots, especially in valleys and canyons where pollutants tend to accumulate. Conversely, DACNO<sub>2</sub> provides lower NO<sub>2</sub> estimates in areas between the mountains with few sources. Overall, DACNO<sub>2</sub> provides more spatially detailed 3D NO<sub>2</sub> fields, revealing greater variability in the vertical profiles across different grids in this complex terrain. This refinement is important, as small point and line sources can contribute significant NO<sub>2</sub> in mountainous regions (Kim et al., 2021).



## 4.3 Implications for Satellite NO<sub>2</sub> Retrievals

Figure 7. Impact of a-priori profile selection on TROPOMI tropospheric NO<sub>2</sub> column retrievals for 2023. (a) Annual mean TROPOMI NO<sub>2</sub> columns retrieved using the original TM5 (about 100 km), CAMS-S5P (10 km), and DACNO<sub>2</sub>-S5P (2 km) a-priori profiles. (b) Spatial distribution of the relative difference (%) in TROPOMI NO<sub>2</sub> columns retrieved with three profiles. (c) The relative change in retrieved NO<sub>2</sub> columns across three subregions (the northern region, Paris, and the Alpine region) when using DACNO<sub>2</sub>-S5P versus CAMS-S5P profiles.

To assess the potential of DACNO<sub>2</sub> for satellite NO<sub>2</sub> product improvement and development, we tested its use as a source of a-priori NO<sub>2</sub> profiles in TROPOMI retrievals. For this, a dedicated version of the model (DACNO<sub>2</sub>-S5P) was developed for the TROPOMI overpass time, predicting a 3-hour average NO<sub>2</sub> (11:00–13:00 UTC) using the same three-phase strategy. The model targets, named CAMS-S5P and EEA-S5P, represent process-based and measured NO<sub>2</sub> data during this period.





Model evaluation (Table S1) shows DACNO<sub>2</sub>-S5P agrees well with CAMS-S5P (RMSE =  $0.98 \mu g/m^3$ , r = 0.94,  $R^2 = 0.88$ , bias =  $0.03 \mu g/m^3$ ) at 10 km grids. Compared to EEA-S5P measurements, DACNO<sub>2</sub>-S5P achieves better agreement (RMSE =  $5.07 \mu g/m^3$ , r = 0.77,  $R^2 = 0.59$ , bias =  $0.05 \mu g/m^3$ ) than CAMS-S5P (RMSE =  $5.27 \mu g/m^3$ , r = 0.76,  $R^2 = 0.55$ , bias =  $-0.94 \mu g/m^3$ ).

We replaced the original TM5 a-priori profiles (about 100 km resolution) in the TROPOMI retrievals with CAMS-S5P and DACNO<sub>2</sub>-S5P profiles, following the approach described in Douros et al. (2023) and focusing on the troposphere. Fig. 7a presents the annual mean TROPOMI NO<sub>2</sub> columns retrieved using these different a priori profiles, with inter-comparisons shown in Figs. 7b and 7c. Both CAMS-S5P and DACNO<sub>2</sub>-S5P profiles lead to substantial increases in the retrieved NO<sub>2</sub> columns, by 36.2% and 39.8% on average, respectively. The increase associated with CAMS-S5P is consistent with previous findings (Douros et al., 2023) and is primarily attributable to the improved spatial resolution of the a-priori profile, which enhances the sensitivity of TROPOMI retrievals to NO<sub>2</sub> hotspots (Tack et al., 2021; Ialongo et al., 2020).

Compared to the CAMS-S5P profile, using DACNO<sub>2</sub>-S5P as the a priori increases retrieved NO<sub>2</sub> columns by about 3.0% on average (Fig. 7b), associated with the reduced negative bias against EEA-S5P measurements reported above. In central-western France (0°E–2.6°E, 45.6°N–46.3°N), a distinct southwest-to-northeast line of reinforced NO<sub>2</sub> columns appears because DACNO<sub>2</sub>-S5P enhances the emission signals from the cities of Angoulême, Guéret, and Montluçon. Regional comparisons (Fig. 7c) show that the DACNO<sub>2</sub>-S5P profile leads to a 1.8% increase in the northern region and 5.9% in Paris, with the most significant increases surrounding major emission hotspots. This is likely due to DACNO<sub>2</sub>'s enhanced ability, based on finer resolution, to capture small-scale emission sources and resolve strong spatial gradients around NO<sub>2</sub> hotspots. In the Alpine region, the average increase reaches 1.7%, ranging from -18.1% (5th percentile) to +24.1% (95th percentile) between the surrounding areas and the central mountains, with a similar pattern observed in the Pyrenees. The absolute difference remains small, ranging from -3.94 × 10<sup>14</sup> molecules/cm<sup>2</sup> (5th percentile) to 5.39 × 10<sup>14</sup> molecules/cm<sup>2</sup> (95th percentile). This large fluctuation reflects the complex NO<sub>2</sub> distribution in the mountainous region and benefits from high-resolution modeling, as DACNO<sub>2</sub> estimates can reach lower background values or enhance the hotspots signal in this region.


These results illustrate the potential of using DACNO<sub>2</sub> profiles to improve satellite NO<sub>2</sub> retrievals, particularly for evolving high-resolution instruments. However, the DACNO<sub>2</sub> product remains a prototype, and a roadmap for operational deployment is outlined in the conclusion.

# 4.4 Generalization Capability and Data Quality: Insights from the COVID-19 Period

Table 3. Performance of DACNO2 on the 2020 dataset.

| Year 2020                | DACNO <sub>2</sub> -Phase-2 |      |                |              | DAC          | DACNO <sub>2</sub> -Phase-3-2020 |                |              |                   | CAMS-2020-2km |                |              |  |
|--------------------------|-----------------------------|------|----------------|--------------|--------------|----------------------------------|----------------|--------------|-------------------|---------------|----------------|--------------|--|
| EEA                      | RMSE (ug/m³)                | r    | $\mathbb{R}^2$ | bias (ug/m³) | RMSE (ug/m³) | r                                | $\mathbb{R}^2$ | bias (ug/m³) | RMSE (ug/m³)      | r             | $\mathbb{R}^2$ | bias (ug/m³) |  |
| Total                    | 5.78                        | 0.80 | 0.63           | -0.27        | 5.47         | 0.82                             | 0.67           | 0.01         | 4.99              | 0.88          | 0.73           | -2.05        |  |
| Urban                    | 6.06                        | 0.81 | 0.63           | -1.53        | 5.62         | 0.83                             | 0.68           | -1.28        | 5.83              | 0.88          | 0.65           | -3.44        |  |
| Suburban                 | 5.81                        | 0.80 | 0.58           | 1.12         | 5.63         | 0.82                             | 0.60           | 1.55         | 3.97              | 0.91          | 0.80           | -1.08        |  |
| Rural                    | 4.97                        | 0.81 | 0.63           | 1.08         | 4.84         | 0.82                             | 0.65           | 1.24         | 3.80              | 0.89          | 0.78           | 0.14         |  |
| DACNO <sub>2</sub> -10km |                             |      |                |              |              |                                  |                |              |                   |               |                |              |  |
| Total levels             | 1.66                        | 0.91 | 0.80           | -0.43        | 1.62         | 0.91                             | 0.81           | -0.34        |                   |               |                |              |  |
| L0                       | 2.18                        | 0.93 | 0.84           | 0.43         | 2.09         | 0.94                             | 0.86           | 0.48         |                   |               |                |              |  |
| L50                      | 1.95                        | 0.93 | 0.84           | 0.57         | 2.06         | 0.94                             | 0.82           | 0.77         |                   |               |                |              |  |
| L250                     | 1.97                        | 0.89 | 0.74           | -0.70        | 1.89         | 0.88                             | 0.76           | -0.50        | G.L.              | . 50. 000     | 0 (101         | ,            |  |
| L500                     | 2.06                        | 0.83 | 0.50           | -1.01        | 1.94         | 0.82                             | 0.55           | -0.81        | CAMS-2020 (10 km) |               |                |              |  |
| L1000                    | 1.64                        | 0.72 | 0.00           | -1.01        | 1.57         | 0.70                             | 0.09           | -0.93        |                   |               |                |              |  |
| L2000                    | 1.42                        | 0.49 | -0.81          | -1.01        | 1.40         | 0.47                             | -0.77          | -0.99        |                   |               |                |              |  |
| L3000                    | 0.81                        | 0.46 | -0.99          | -0.59        | 0.80         | 0.45                             | -0.95          | -0.59        |                   |               |                |              |  |




| L5000 | 0.17 | 0.55 | -0.70 | -0.12 | 0.17 | 0.56 | -0.64 | -0.12 |
|-------|------|------|-------|-------|------|------|-------|-------|
|       |      |      |       |       |      |      |       |       |

Note: Similar to Table 2, but for the year 2020.

As noted in Section 2.4.1, CAMS NO<sub>2</sub> data for 2020 were excluded from the training set based on preliminary experiments showing that their inclusion substantially reduced model generalization at higher levels. Since 2020 was marked by the COVID-19 pandemic and large reductions in anthropogenic emissions (Levelt et al., 2022), we specifically evaluated DACNO<sub>2</sub>'s predictive performance for this atypical year. To this end, the DACNO<sub>2</sub>-Phase-2 model was fine-tuned on 2020 EEA NO<sub>2</sub> data, following the same phased development strategy, to produce DACNO<sub>2</sub>-Phase-3-2020.

Table 3 summarizes the 2020 evaluation results, following the format of Table 2. Both DACNO<sub>2</sub>-Phase-2 and DACNO<sub>2</sub>-Phase-3-2020 reproduced observed surface NO<sub>2</sub> concentrations well (e.g., DACNO<sub>2</sub>-Phase-3-2020: RMSE = 5.47 μg/m³, r = 0.82, R² = 0.67, bias = 0.01 μg/m³), with performance comparable to CAMS (RMSE = 4.99 μg/m³, r = 0.88, R² = 0.73, bias = -2.05 μg/m³) but notably lower bias. This demonstrates robust generalization by DACNO<sub>2</sub> under the emission anomalies of the pandemic year. Agreement between DACNO<sub>2</sub> and CAMS remains strong at low altitudes (e.g., surface: RMSE = 2.09 μg/m³, r = 0.94, R² = 0.86, bias = 0.48 μg/m³), but declines rapidly above 1000 m, where R² values approach zero or become negative, indicating a failure to reproduce high altitude CAMS NO<sub>2</sub> distributions for 2020.

A comparison of CAMS NO<sub>2</sub> vertical distributions from 2019 to 2023 (Fig. S1) shows generally consistent annual patterns, except for 2020, which is characterized by anomalously high values above 1000 m. This anomaly is also noted in the CAMS 2020 annual evaluation report (Meleux et al., 2023), which attributes it to some sub-models producing unexpectedly high NO<sub>2</sub> at high levels, resulting in inflated tropospheric column estimates. The underlying causes remain unresolved and require further investigation. These findings highlight the importance of data screening, such as checking distributions and identifying outliers, before model training. Including biased or anomalous target data can introduce noise, increase the risk of overfitting, and reduce generalization performance.







## 5. Conclusions and outlook

This study presents the Deep Atmospheric Chemistry NO<sub>2</sub> model (DACNO<sub>2</sub>), a deep learning model for daily, high-resolution (2 km) 3D NO<sub>2</sub> estimation. DACNO<sub>2</sub> integrates multi-source and multi-modal input features, including emissions, geography, meteorology, and temporal indicators. It uses a multi-constraint and phased training approach to learn from both process-based CAMS NO<sub>2</sub> and measured EEA NO<sub>2</sub> data. This approach allows DACNO<sub>2</sub> to reproduce broad-scale, process-based NO<sub>2</sub> patterns and capture local NO<sub>2</sub> gradients. Results show that DACNO<sub>2</sub> significantly improves the ability to resolve fine-scale spatial patterns, near-surface NO<sub>2</sub> variability, and vertical distribution. It also generalizes well across different spatial areas (urban, rural, mountainous, and emission hotspot regions) and periods of anomalous emissions. Furthermore, the framework demonstrates transferability and flexibility, allowing the model to be fine-tuned to adapt to future emission scenarios and to be adjusted to produce outputs for specific satellite overpass times in addition to daily averages.

A systematic evaluation shows that DACNO<sub>2</sub> outperforms the state-of-the-art regional CAMS product in reproducing measured surface NO<sub>2</sub> concentrations. Overall, DACNO<sub>2</sub> achieves a lower RMSE (4.99 vs. 5.32  $\mu g/m^3$ ), higher correlation (r = 0.82 vs. 0.80, R<sup>2</sup> = 0.66 vs. 0.61), and a substantially reduced bias (-0.38 vs. -1.15  $\mu g/m^3$ ). The improvement is particularly significant in urban areas with high spatial variability, where DACNO<sub>2</sub> yields a higher R<sup>2</sup> (0.64 vs. 0.56) and halves the negative bias (-1.42 vs. -2.89  $\mu g/m^3$ ). In rural areas with very low background levels, DACNO<sub>2</sub> also shows a better R<sup>2</sup> (0.55 vs. 0.51) and maintains a small positive bias, in contrast to the large one from CAMS (1.05 vs. 1.98  $\mu g/m^3$ ). Vertical profile analysis indicates that DACNO<sub>2</sub> provides greater spatial detail and variation than CAMS, capturing small-scale emission sources and topographic influences more effectively. Feature importance analysis highlights the contribution of high-resolution emission proxies, land cover, and multi-level meteorological information to resolving spatial and vertical NO<sub>2</sub> patterns. In contrast, single-level meteorological variables contribute minimally, likely because part of their information is already captured by the more comprehensive multi-level data. This redundancy suggests opportunities for future model optimization.

Application to satellite NO<sub>2</sub> retrievals demonstrates that using DACNO<sub>2</sub>-generated a-priori profiles enhances the sensitivity of TROPOMI NO<sub>2</sub> products to near-surface concentrations and emission hotspots, particularly in small-scale emission sources and complex geographic regions. These findings underscore the potential of high-resolution ML-based profiles for future high-resolution satellite retrievals. However, DACNO<sub>2</sub> remains a prototype, and further work is needed for operational deployment. First, this would involve extending the model's




output to continuous hourly profiles over a broader geographic domain. Second, the model would need to be operated on a robust GPU computational platform with automated data pipelines. Third, a routine validation framework would need to be established to continuously monitor performance against various data, such as CAMS NO<sub>2</sub>, EEA NO<sub>2</sub>, and vertical measurements (e.g., MAX-DOAS). Finally, this operational system would require a strategy for periodic model fine-tuning to adapt to evolving emission patterns and maintain long-term accuracy.

Analysis of model performance during COVID-19 indicates that DACNO<sub>2</sub> consistently generalizes well despite emission anomalies. The inconsistencies observed in CAMS reanalysis for 2020 at high levels highlight the need for screening and quality assurance in model training data to avoid learning biased patterns and degrading model reliability.

Future development of DACNO<sub>2</sub> may include integrating high-resolution 3D process-based NO<sub>2</sub> fields from models such as WRF-Chem and column observations from satellites to strengthen constraints at higher altitudes (e.g., above 2000 m), exploring transformer architectures for improved scalability and multi-modal data processing, and embedding additional physical constraints into the loss function. Extension to continental or global applications (including data-poor regions such as the African continent) will further support large-scale air quality management and atmospheric chemistry research.

## Data and code availability

The daily number of flights is accessible at <a href="https://www.eurocontrol.int/Economics/DailyTrafficVariation-States.html">https://www.eurocontrol.int/Economics/DailyTrafficVariation-States.html</a>. The CAMS global emission inventories are accessible at <a href="https://ads.atmosphere.copernicus.eu/">https://ads.atmosphere.copernicus.eu/</a>. The GRIP global roads database can be downloaded from <a href="https://www.globio.info/download-grip-dataset">https://www.globio.info/download-grip-dataset</a>. The VIIRS nighttime light data can be accessed from <a href="https://eogdata.mines.edu/products/vnl/">https://eogdata.mines.edu/products/vnl/</a>. The population dataset is provided by <a href="https://ec.europa.eu/eurostat/web/gisco/geodata/population-distribution/population-grids">https://ec.europa.eu/eurostat/web/gisco/geodata/population-distribution/population-grids</a>. The MERIT DEM data is accessible via <a href="https://eda.dimer.org/">https://eda.dimer.org/<a href="https://eda.dimer.org/">https://eda.dimer.org/<a href="https://eda.dimer.org/">https://eda.dimer.org/<a href="https://eda.dimer.org/">https://eda.dimer.org/<a href="https://eda.dimer.org/">https://eda.dimer.org/<a href="https://eda.dimer.org/">https://eda.dimer.org/<a href="https://eda.dimer.org/">https://eda.dimer.org/<a href="https://eda.climate.copernicus.eu/">https://eda.climate.copernicus.eu/</a>. The CAMS European air quality reanalyses dataset is accessible via <a href="https://eda.atmosphere.copernicus.eu/">https://eda.atmosphere.copernicus.eu/</a>. The

https://doi.org/10.5194/egusphere-2025-4259 Preprint. Discussion started: 24 November 2025

© Author(s) 2025. CC BY 4.0 License.



EGUsphere Preprint repository

EEA AirBase dataset can be downloaded from <a href="https://eeadmz1-downloads-webapp.azurewebsites.net/">https://eeadmz1-downloads-webapp.azurewebsites.net/</a>. The official TROPOMI NO2 product is accessible via the Copernicus Data Space Ecosystem (<a href="https://dataspace.copernicus.eu/">https://dataspace.copernicus.eu/</a>). The data generated for this study can be accessed from the Zenodo data archive

(Sun et al. (2025), <a href="https://doi.org/10.5281/zenodo.16986854">https://doi.org/10.5281/zenodo.16986854</a>).

The DACNO<sub>2</sub> model and its framework are built using the Pytorch library (<a href="https://pytorch.org/">https://pytorch.org/</a>) in the Python environment. All code related to model design and data processing is available upon request from the corresponding author.

655 Author contribution

WS, FT, and MVR conceived the study. WS built the model, performed all analyses, and wrote the initial draft of the manuscript. FT, LC, and MVR reviewed and revised the draft. All authors substantially contributed to the final manuscript.

**Competing interests** 

The co-author, MVR, is a member of the editorial board of Atmospheric Chemistry and Physics. The other authors declare no competing interests.

Acknowledgements

The Belgian Federal Science Policy Office is gratefully appreciated for funding part of this work in the framework of the Terrascope-S5P PRODEX project (PEA 4000136290); L.C. is a research associate supported by the Belgian F.R.S.-FNRS. We used AI-assisted tools to polish the manuscript. The authors are solely responsible for the scientific content and interpretations.

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
