# Peer review of "Technical Note: DACNO2 – A Multi-Constraint Deep Learning Framework for High-Resolution 3D NO2 Field Estimation"

_EGUsphere, 2025_

## Referee Comment (RC1)

**Review on Technical Note: DACNO2 – A Multi-Constraint Deep Learning Framework for High-Resolution 3D NO2 Field Estimation**

In this manuscript, Sun et al. present a new machine learning (ML) model that estimates daily average 3D distributions of NO2 in central Europe. The model is trained on data from the CAMS ensemble CTM and in situ observations at the surface. Because NO2 plays an important role in atmospheric chemistry and air pollution, and the model proposed by Sun et al. outperform CAMS, the results of this manuscript are definitely worth publishing and are well in scope of ACP.

Before listing my comments below, I would like to emphasize the strengths of the manuscript in its current form. The manuscript contains several clever ideas, e.g. using flight data as a proxy for exceptional periods, such as holidays. It is apparent from the detailed explanations of the training procedure that the presented work was overall well conceptualized and executed. The model is quite complex and represents a significant body of work. The model's evaluation is detailed.

I recommend to address the following comments.

**General comments**

- 1) Why is the manuscript submitted as a technical note? The novelty and quality of the proposed model is surely sufficient for this manuscript to be a regular research article at ACP.
- 2) It is interesting to see that satellite data were deliberately *not* used as input to DACNO2. Nonetheless, satellite data would have resolved a current weak point of the model by capturing the temporal variability of the NO2 burden. Currently, DACNO2 appears to use static emission data (average of the year 2018). Looking at the other input data from Table 1, I cannot identify any input variable from which the model could determine trends across seasons and years. Re-training the model (which apparently took > 3 weeks) is surely too much to ask, but the fact that DACNO2 is essentially barren of any temporal dynamic on the seasonal/yearly scale should be discussed.
- 3) A significant drawback of training on CAMS data is that they cannot account for one of the most significant measurement biases of in situ instrument: The NOy bias of molybdenum-based photolysis converters that many instruments in Europe use. This has been discussed elsewhere, see e.g. Lamsal et al. (2008) and Villena et al. (2012). Although the problem cannot be resolved based on CAMS data, it should be acknowledged in the paper.
- 4) In many places the manuscript assumes a fair amount of ML terminology/knowledge from the reader. Terms such as LSTM, inception modules, latent space, residual connections, max-pooling branch, dropout, and batch norm are often used with very brief, or no references/explanations. I am afraid that non-ML readers might be overwhelmed by this.
- 5) Some parts of sect. 2.2–3.3 are very detailed in their technical elaborations, but still leave questions wrt. the broader methodology. For example, sect. 2.2.3: I am sure that the readers will understand *what* was done, but not necessarily *why*. Why is it necessary for the patches to overlap? Why 12 patches per day? How does all of this help to "balance the model's receptive field and computational efficiency", as the authors claim?
- 6) I do not think it is correct to say that the use of a priori profiles enhances the *sensitivity* of the satellite measurements (I. 55). Their purpose is rather to "fill" the gap of missing NO2 that arises from a lack of measurement sensitivity. It should also be mentioned that the true NO2 profiles extend beyond 5 km, and often show raised concentrations at 8000–12000 m, see e.g. Douros et al. (2023). On one hand, the effect of this "tail" on the resulting NO2 VCD is expected to be rather small. On the other hand, the difference of DACNO2-S5P to TROPOMI CAMS is only a few percent, hence the authors are arguably in a territory where the discussion of small effects is relevant.
- 7) I would recommend writing out (horizontal) spatial resolutions more carefully. Consider writing  $_{,2}$  km $_{,2}$

**Specific comments**

- I. 19: To find "physically interpretable relationships" is a very high standard, and the manuscript does not come back to this anywhere; consider lowering the expectations a little bit.
- I. 64: what is the difference between "NO2 fields generated by CTMs", and "process-based CTM outputs"?
- I. 178: Multiplication by an inverse = division.
- I. 311: The best model out of what ensemble? This suggests that the authors tested multiple variants of the model, but the differences between the variants are not explained.
- sect. 4.1: I would suggest to shorten the discussion on the feature importance, because Fig. 5 already speaks for itself.

**References**

Lamsal et al. (2008). Ground-level nitrogen dioxide concentrations inferred from the satellite-borne Ozone Monitoring Instrument. Journal of Geophysical Research: Atmospheres, 113(D16).

Villena et al. (2012). Interferences of commercial NO2 instruments in the urban atmosphere and in a smog chamber. Atmospheric Measurement Techniques, 5(1):149–159.